# NEGATION HANDLING FOR AMHARIC SENTIMENT CLASSIFICATION

## ABSTRACT

User generated content contains opinionated texts not only in dominant languages (like English) but also less dominant languages( like Amharic). However, negation handling techniques that supports for sentiment detection is not developed in such less dominant language(i.e. Amharic). Negation handling is one of the challenging tasks for sentiment classification. Thus, this work builds negation handling schemes which enhances Amharic Sentiment classification. The proposed Negation Handling framework combines the lexicon based approach and character ngram based machine learning model. The performance of framework is evaluated using the annotated Amharic News Comments. The system is outperforming the best of all models and the baselines by an accuracy of 98.0. The result is compared with the baselines (without negation handling and word level ngram model).
**keywords:Negation Handling Algorithm, Amharic Sentiment Analysis, Amharic Sentiment lexicon, char level, word level ngram, machine learning, hybrid**

## 1 INTRODUCTION

Due to the emergence of social media including Facebook, Twitter, YouTube, Insta-grams, LinkedIn and so on, the number of users who participates and consumes in-formation is increasing very fast. Users usually express their feelings, emotions and opinions as comments in response to the posted news, photo, audio and video. The texts in social media are informal as it contains spelling errors, slangs, abbreviations or users might use different language. Preprocessing these unstructured and informal texts is very challenging prior to sentiment analysis. Preprocessing includes tokeniza-tion, punctuation mark removal, stop word removal, abbreviation expansion, language detection, spelling error handling, normalization and stemming. The preprocessing task could be harder for under resource languages (e.g. Amharic). Sentiment Analysis is the process of framing the unstructured texts to detect, extract and classify opinion words. Currently, opinionated sources are increasing in languages other than English. Amharic is one of these resource-limited languages. However, Amharic sentiment analysis researches are very few as it has no sufficient linguistic resources for linguistic preprocessing and sentiment analysis. There are several challenges in lexicon based sentiment analysis. One of these is that handling negation in the text. The most common approach for negation handling is carried out relying on negation keywords. However, it is challenging to identify the scope of negation where the process of correctly identifying the part of the text affected by the presence of negation word. In most negation handling researches, simple approach of detection of negation scope is used. Amharic is one of the most spoken Semitic languages in the world next to Arabic. Amharic is morphologically rich and this adds another challenge to detect and extract opinion words in sentiment analysis. Inspired by the aforementioned preprocessing problems, negation handling is never studied in Amharic language to the best of our knowledge. Thus, the main aim of this research is develop an automatic method to handle negation and combined with char ngram features for Amharic Sentiment Analysis relying on these linguistic features. The research questions to be addressed in this work are as follows: (a) how can we automatically detect negation words in Amharic texts?, (b) how can we design a framework for handling negation in Amharic sentiment analysis?, (c) how to capture char level ngram features for improving Amharic Sentiment Analysis in Social media(e..g. Facebook) and (d) how can we evaluate the performance of the framework? The rest of this paper is organized as follows: in the section 2, the related works are presented. The proposed approaches including negation handling and capturing char level ngram features for Amharic Sentiment Analysis is described in section 3. In section 4, results and discussions are presented.

Conclusions and recommendations are drawn in section 5. In this section, we briefly present the key related works. Asmi & Ishaya (2012) proposed an approach that can detect negation and considers scope of negation relying on syntactic dependency for sentiment analysis. In (Amalia et al., 2018), proposed rule based negation handling and its scope based on syntactic parsing of Indonesian language for machine learning based sentiment analysis in twitter and the result of support vector machine performs well as compared with other experiments. The F-Score on two Twitter data sets is improved by 1.79% and 2.69% from the existing baseline without negation handling. In (Farooq et al., 2017), develop negation scope handling strategies relying on the linguistic features of the language. The accuracy of negation scope identification is 83.3% which outperforms very well above the baseline. In (Diamantini et al., 2016), develops negation scope detection relying on dependency parsing tree and semantic disambiguation technique and evaluated on integrated social networks in real time. The performance of negation handling is outperforming with accuracy of 6% more than the baseline. The work in (Heerschop et al., 2011) develops negation handling relying on wordbank creation and document sentiment scoring for sentiment analysis and outperforms the human rating by an increase precision with 1.17%. In (Enger et al., 2017), developed negation cue and scope handling open source tool relying on dependency parser, negation cue(negation lists, prefix, suffix) as input to machine learning(e.g. SVM) and its performance is slightly unchanged from the baseline.

## 2 METHODS

### 2.1 DATA SETS AND LEXICAL RESOURCES

This subsection describes the main data sets and lexical resources used for building and evaluating the proposed framework.

### 2.2 FACEBOOK ANNOTATED AMHARIC NEWS COMMENTS

This dataset consists of 2705 sentence/phrase level sentiment annotated facebook news users' comments collected from the Government Office Affairs Communication (GOAC) between 2008 and 2010. News that received high view counters / frequent comments were selected as "hot topics" and the associated comments labeled by professionals into either positive or negative sentiment. The data sets for evaluating the performance of the negation handling and char level ngram based model for Amharic sentiment analysis relying on sentence level sentiment scoring.

### 2.3 AMHARIC SENTIMENT LEXICONS

The Amharic sentiment lexicons includes manual(1000) (Gebremeskel, 2010), SWN(13679) (Neshir Alemneh et al., 2019) and SOCAL(5683) (Neshir Alemneh et al., 2019) .

### 2.4 PROPOSED APPROACHES

The proposed framework includes the algorithm for negation handling and the sentiment scoring strategies. The framework for negation handling and char level ngram based model is shown in Figure 1 below.

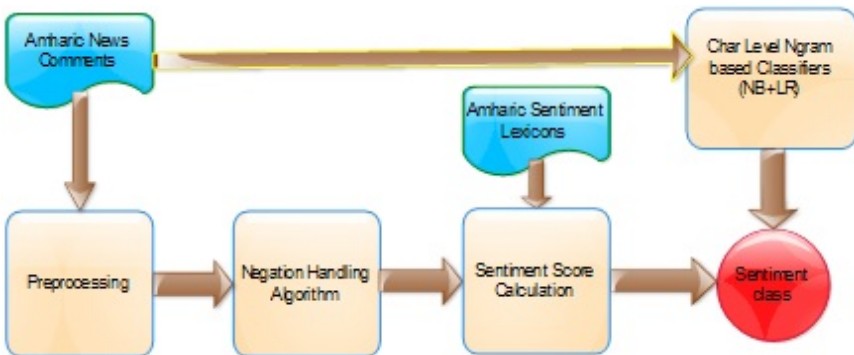

Figure 1: Proposed Framework for Amharic Sentiment Classification

The stages in the framework are briefly described as follows:

## 2.5 Preprocessing

We apply basic preprocessing on Amharic News Comments. These include normalization of Amharic script symbols, tokenization, stop word removal, punctuation mark removal and so on. Amharic writing system is expressed using only consonants. To handle the features of the language is very challenging. We require conversion of Amharic scripts to consonant-vowel form. Particularly, before performing negation handling and stemming, the algorithm converts each Amharic word to its consonant vowel form.

## 2.6 Negation Detection

We developed negation detection algorithm which re-turns true if a word contains negation cue either prefix(ዕአልኈምኽ, ስልኈምአይ, ዕኈትአ, ብአይ, ምኣይ, ዕኈል, ዕኈት, ዕኈን, ዕኈይ, ይኣል, ስአተ, ስኣን, ስአል, ስአይ) , suffix (ብኢ.ስ, ይኈልኈኸ) or a word in negation lists. Before a word is checked for negation clue, the Amharic word should be converted to consonant – vowel form. The aforementioned Amharic prefixes and suffixes indicating negations which are written in consonant- vowel form. Figure 2 represents the finite state automata of Amharic morphological negation clues indicated by using prefix, suffix or negation carrying word. An Amharic word is given as input to this finite state automaton and the finite state automata (FSA) process the patterns of the word relying on the regular expressions. If the word contains either of the prefixes, it will finish processing on the state containing stem. If it contains one of the suffixes, it will finish processing on the finish state or if it contains neither prefix nor suffix, it will finish on the start state. And then check the word is in the negation list(NL). If any of these is satisfied, it will return true, otherwise returns false.

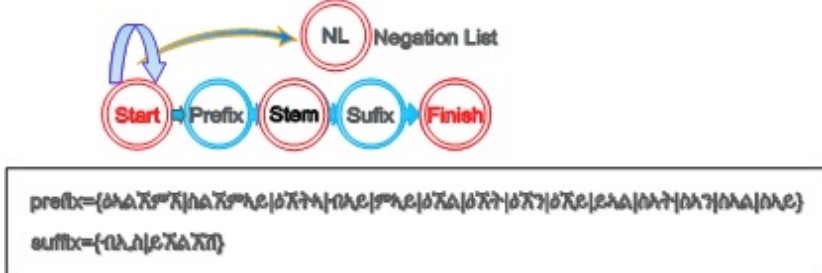

Figure 2: State Automata for Amharic Morphological Negation Clues

Assume the $i^{th}$ Amharic news comment, $C_i$ in Amharic News Comments collection, da, has preprocessed and finally, the comment is tokenized into lists. That is, $C_i = w_{i1}, w_{i2}, ..., w_{ij}, ..., w_{iN}$, ), where N is the number of tokens in $i^{th}$ comments, $C_i$. Let the Amharic Sentiment Lexicon is denoted by $S_a$. As part of preprocessing, we normalized not only all Amharic words in the Amharic News Comments but also handling entries of Amharic Sentiment Lexicon by replacing varied alphabets of the same sound with identical symbols. Moreover, a stemmer is applied for after negation identification is completed. As Amharic is morphologically rich, to reduce the mismatch of Amharic words during string comparison operation, we used stemming for this purpose. Below is the Negation handling algorithm in 1.

**Input:** $word_{orginal}$: input word
**Output:** $Is\_nagation$: returns True if there is negation clue, otherwise false

1 sufix← r'(.*?){ (ብኢ.ስ|ይኸልኸሸ)}?$'   prefix← r̂'(.*?){ r̂'(ዕኣልኸምኸ|ስልኸምኣይ|ዕኸትኣ|ብኣይ|ምኣይ|ዕኸል|ዕኸት|ዕኸን|ዕኸይ|ይኣል|ስኣት|ẻ
    } word_CV_form←convert_to_CV($word\_original$) prefix_true*gets*False suffix_true*gets*False

2 **if** *len(word)>3:* **then**

3 | stem, suffix *gets* re.findall(negation_suffix, word_CV_form)[0]    word*gets* stem

4 **if** *len(suffix)>1:* **then**

5 | suffix_true*gets*True    try: stem,prefix *gets* re.findall(negation_prefix, stem)[0][1], re.find-
    all(negation_prefix, stem)[0]    word*gets* stem    **if** *llen(prefix)>1:* **then**

6 | | prefix_true*gets* True    except: pass

7 **if** *prefix_true:* **then**

8 | Return prefix_true;

9 **if** *suffix_true:* **then**

10 | Retrun suffix_true

11 **if** *suffix_true is False and prefix_true is False* **then**

12 | Return False

algorithm 1: Amharic Negation Handling Algorithm

## 2.7 SENTIMENT SCORE CALCULATION

For each Amharic news comment, $C_i$, if each stemmed word $w_{ij}$ is found in either of the Amharic Sentiment lexicons (Manual, SOCAL, SWN), then the sentiment score $s_{ij}$ is retrieved. $s_{ij}$ and its position index in the comment is stored. To compute the sentiment of the comment, we apply positional weighting inversion if the comment contains any negation clue. If negation clue is not found, the score of the word is simply added. The sentiment Score computing algorithm is presented in 2.

**Input:** $Input$: The $i_{th}$ Amharic News Comment, $C_i$ of the comment Collection.
**Output:** $Output$: SentimentClass(either positive, negative or neutral/mixed)

13 $S_a$← Load Amharic Sentiment Lexicons from file   Initialize parameters: $Score_Index ←, Senti_score = 0, Negation = 0$   $C_i ← [wi1, wi2, ..., wij, ..., wiN] = preprocess(C_i)$, tokenize, normalize the comment

14 value*gets* False

15 **foreach** $w_{ij}$ *in* $C_i$ **do**

16 | **if** $w_{ij}$ *in* $S_a$ **then**

17 | | Score_Index[j] ← $S_a[w_{ij}]$

18 | **if** *DetectNegation($w_{ij}$):* **then**

19 | | Negation← j

20 | **if** *Negation is not zero:* **then**

21 | | **foreach** *k in Score_Index.keys:* **do**

22 | | | **if** *k < Negation and Negation is not Zero:* **then**

23 | | | | Senti_Score=k* Senti_Index[k]  Sum=Sum+k

24 | | | **if** *k>Negation and Negation is not Zero:* **then**

25 | | | | N=len(Senti_Index.keys) Senti_Score=Negation*(N-k)* Senti_Index[k]  Sum=Sum+(N-k)

26 | **if** *Negation is zero:* **then**

27 | | N=len(Senti_Index.keys()) Senti_Score=sum(Senti_Index.Values)/N

28 | | **if** *(Sum>0:* **then**

29 | | | Senti_Score=Senti_Score/Sum

30 **if** *Senti_Score>0:* **then**

31 | Return "Positive"

32 **if** *SentiScore<0:* **then**

33 | Return "Negative"

34 **if** *SentiScore==0:* **then**

35 | Return "Neutral/Mixed/Unclassified"

algorithm 2: Amharic Sentiment Score Calculation Algorithm

As specified in algorithm 2, the sentiment class of the comment is decided based on the value of the computed sentiment score. If the score is greater than zero, then the sentiment of the comment is positive. If the score is less than zero, the sentiment of the comment is negative. Otherwise, the sentiment of the comment is unclassified or neutral or mixed.

## 2.8 Char Level Ngrams for Amharic Sentiment Analysis

In this stage, we propose machine learning based Amharic Sentiment classification that takes the character level ngrams tfidf features of Amharic text. Yet, Amharic language lacks standard linguistic specific tools including dependency parser and part of speech tagger to capture the scope of negation in Amharic text for sentiment Analysis. Besides lexicon based negation handling approach, we try to enhance the performance of Amharic sentiment Analysis relying on char level ngram tfidf feature weights. As we hoped that there is flexibility to capture language dependent features using character level ngram models. The usefulness of character aware language models is well suited to apply for language identification, reducing of text feature sparse dimensionality, helps to handle spelling errors, abbreviations, special characters and so on [7]. These problems are particularly found in texts of social media where texts data are too noisy and informal in nature to be processed using standard natural language processing toolkit. That is why we propose character level ngram approaches to reduce and address these issues for Amharic facebook news comments' sentiment analysis rather than word level ngram approaches. For example, the negation carrying Amharic word "አልወደውም"/ "I donot like him"/ has 2-gram character level features includes: አ-አል-ልወ-ወደ-ደው-ውም-ም, 3-gram character level features are አ- አል-አልወ-ልወደ-ወደው-ደውም-ውም-ም and so on. In both cases, the negation marker/morpheme/ አል- is detected as feature of the negation word/"አልወደውም"/. Thus, we can identify the negation word in Amharic text relying on character level ngram features and in turn its scope will also be captured in the same manner from its surrounding character ngram features that are assumed to enhance the performance of sentiment analysis. For each Amharic facebook news comment, term frequency-inverse document frequency (tfidf) weighted char level ngrams(e.g. bigrams and trigrams) features are extracted and becomes input to ma-chine learning classifiers (e.g. logistic regression and Naïve Bayesian) for training and validation. Then, the two machine learning algorithms are ensembled with lexicon based negation handling algorithm to predict the sentiment class of Amharic comments. The usefulness of the combined (hybrid) framework is measured using testing data set partitioned from annotated Amharic news comments. The framework is implemented using python scikit learn library and the result and discussion are presented in the following subsection.

## 3 Results and Discussions

In this section, we will evaluate the usefulness of the combination of Negation Handling(NH) and character level ngram based models for sentiment classification of Amharic facebook news comments. We call the combination of rule based NH and char ngram based models as hybrid model for Amharic Sentiment classification. In negation handling approach, Amharic texts require basic text preprocessing (tokenization, punctuation mark removal, normalizing Amharic script symbols, stopword removal, spelling corrector, stemming, etc.). The effect of stemming and negation detection technique on Amharic text is investigated to increase the accuracy of lexicon based Amharic sentiment classification. However, to extract char level ngram features, pre-processing tools including stemming, removal of punctuation marks, removal of stop words, abbreviation expansion, etc. are not required. Thus, prior to char ngram based sentiment classification, we partition the Annotated Amharic Facebook News Comment corpus into training and testing sets. Logistic regression(LR) and Naïve Bayesian(NB) models are built relying on the char level bi-gram and tri-gram features of training set. For Amharic Sentiment Classification, the accuracy of the combined models on the test set are presented in Table 1.

Table 1: The Accuracy (in percent) of Amharic Sentiment Classification Models

| Approach | Accuracy(%) | |
|---|---|---|
| | Wordlevel (Baseline) | Char Level |
| Negation Handling(NH) | 86.2 | - |
| NB + LR | 79.32 | 83.75 |
| Hybrid(NH+NB+LR) | 95.27 | 98.0 |

Discussions: In Table 1, the usefulness of the proposed approach is tested relying on their accuracies of classifying sentiment of Amharic facebook news comments. The results in Table 1 show that negation handling algorithm outperforms very well (acc. 86.2%) than the performance of character level and word level based machine learning models for classifying sentiment of Amharic texts. On the other hand, character level ngram based classifier is more useful for classifying Amharic Sentiment than word level ngram models (baseline). Finally, the hybrid model is obtained by combining negation handling approach and char ngram models (NB + LR). This hybrid model outperforms with accuracy of 98% than the other models and its combinations. Yet, it is quite difficult to find why errors are generated in predicting sentiment category of Amharic Facebook News Comment Text. On the basis of on-spot detection, the associated reasons for most of wrongly predicted Amharic News Comments are presented as follows: (a) we detect errors coming from professionals who annotated Amharic News comments. For example, 'የሀገሪቱን ሰላም እያደፈረሱ ያሉት የኢሳት ጋዜጠኞች ናቸው፤ '/ESAT journalists are making the country to be disturbed/. The human annotator assigned positive sentiment label to this comment. It should rather be assigned negative label. To this end, the data should be revisited for correction. (b) we detect some comments contains opinion words but the sentiment orientation of the whole comment is opposite to the polarity of the opinion words in the comment. The reason for this is that the comment is conditional sentence. It talks about hope rather than opinion. For example, the Amharic news comments 'እውነታ የሆነ ነገር መገለፅ ቢችል ጥሩ ነው፤'/It will be good if you tell the truth/. In this comment, there are two positive words (e.g. good and truth). However, the interpretation of the whole comment is talking about that they told us fake news. Similarly, we detect some comments represents wishes that need to be done. For example, 'በቃል የሚነገሩ ነገሮችን በተግባር እንዲፈፀሙልን እንፈልጋለን፤ '/We need to see accomplished in practice that we heard in words/. This comment has no any opinion words which helps the proposed approach to predict the sentiment of the comment correctly. Further researches needs to carry out to reduce the source of errors in predicting sentiment class of Amharic comments. Our recent findings is a good starting point to improve the performance of Amharic sentiment analysis in facebook news comments. Fine tuning char ngram features shows suitedness and flexible for sentiment analysis of resource limited language (e.g. Amharic) than word level ngram models.

## 4 CONCLUSIONS

In general, extensive linguistic resources are expensive to build sentiment analysis on the less dominant languages (e.g. Amharic). To reduce this problem, we propose negation handling approach and char ngram approach for Sentiment analysis of Amharic facebook news comments. The proposed approaches are evaluated by measuring accuracy of individual and their combinations for Amharic text sentiment classification. This work reveals that combining rule based (negation handling approach) with machine learning approach (char level ngram feature models) outperformed the best of the individual approaches. This enhances performance of sentiment classification on Amharic news comments on social media (e.g. Facebook). Some of the contributions of this work are briefly summarized below:

- The proposed negation handling approach for Amharic Sentiment Analysis outperforms very well when we compared to the char ngram based machine learning classifiers.
- The approach for negation handling can be adapted to develop Amharic text stemmer algorithm.

- We developed an approach that can be adapted to sentiment analysis of other resource limited languages.
- The char ngram based machine learning is promising that it reduces the demand for linguistic resources for less dominant languages.
- The approach can also be adapted to other tasks of natural language processing including language identification, topic based text categorization, authorship attributions, just to name a few.
- The hybrid approach (NH + LR + NB) outperforms the best compared to the individual approach for sentiment classification on Amharic facebook news comment texts.
- The code and related resources will be accessible publicly for research communities.

So far, we have seen that the proposed approach still lacks accuracy for sentiment analysis of a specific language (e.g. Amharic). The approach potentially does not sufficiently capture the language specific features that help to identify the sentiment class of Amharic news comment text in social media. Further work should be performed to reduce the amount of errors in sentiment analysis of Amharic facebook news comments. To address these issues, we may need to consider char ngram embedding features from corpus of the same domain(e.g. Facebook news comments).

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
