# OpenReview forum: "Amharic Negation Handling"
_ICLR.cc/2020/Conference — Reject_

### Official Review · AnonReviewer1 · 2019-10-25
**Official Blind Review #1**

**Rating:** 1

**Review:**

This paper is quite difficult to read. The figures are pixelated. There is almost no organization to the text of the work.

Descriptions are imprecise and lax: for example, "We apply basic preprocessing on Amharic News Comments. These include normalization of Amharic script symbols, tokenization, stop word removal, punctuation mark removal and so on. Amharic writing system is expressed using only consonants. To handle the features of the language is very challenging. We require conversion of Amharic scripts to consonant-vowel form. Particularly, before performing negation handling and stemming, the algorithm converts each Amharic word to its consonant vowel form." What kind of normalization? What kind of tokenization? Did you use outside tools? If so, cite them or refer to the code. If not, then please explain your method more thoroughly. Don't assume all your readers know what consonant-vowel form is. "and so on" is not appropriate. You should explain what the so on is in a paper submitted to a conference. Don't bother telling us the features of the language are challenging. Just tell us how you did it.

Switching between words like "cue" and "clue"

Even the font of the text changes part way through the paper.

Figure 2 is almost entirely uninformative.

The algorithms are poorly displayed and nearly unreadable.

Experiments are too limited.

**Experience Assessment:**

I have published in this field for several years.

**Review Assessment: Checking Correctness Of Derivations And Theory:**

N/A

**Review Assessment: Checking Correctness Of Experiments:**

I carefully checked the experiments.

**Review Assessment: Thoroughness In Paper Reading:**

N/A

---

### Official Review · AnonReviewer2 · 2019-10-28
**Official Blind Review #2**

**Rating:** 1

**Review:**

This paper introduces a method combining the lexicon-based feature and character ngram model for handling negation in Amharic sentiment classification. The algorithmic contribution is demonstrated in a Amharic dataset.

My rating for this paper is Reject because there is no novelty in the approach, it introduces some rules to build a FSA plus a logistic regression model with character-level ngram feature. It is also lack of argument/motivation why the proposed model is special and specific to the problem. The experiment only compares with two baselines in single dataset, which is also not convincing.


**Experience Assessment:**

I have read many papers in this area.

**Review Assessment: Checking Correctness Of Derivations And Theory:**

I assessed the sensibility of the derivations and theory.

**Review Assessment: Checking Correctness Of Experiments:**

I assessed the sensibility of the experiments.

**Review Assessment: Thoroughness In Paper Reading:**

I read the paper at least twice and used my best judgement in assessing the paper.

---

### Official Review · AnonReviewer3 · 2019-11-06
**Official Blind Review #3**

**Rating:** 1

**Review:**

While this paper tackles an interesting problem. The technical approach is unfortunately too outdated and obvious and not quite the level of ICLR.
The dataset is likely too easy given the high accuracy.


**Experience Assessment:**

I have published in this field for several years.

**Review Assessment: Checking Correctness Of Derivations And Theory:**

N/A

**Review Assessment: Checking Correctness Of Experiments:**

I assessed the sensibility of the experiments.

**Review Assessment: Thoroughness In Paper Reading:**

I made a quick assessment of this paper.

---

### Decision · Program_Chairs · 2019-12-19

**Decision:**

Reject

**Comment:**

Main content:

This paper presents negation handling approaches for Amharic sentiment classification.

--

Discussion:

All reviewers agree the paper is poorly written, uses outdated approaches, and requires better organization and formatting.

--

Recommendation and justification:

This paper after more work might be better submitted in an NLP workshop on low resource languages, rather than ICLR which is more focused on new machine learning methods.